# Significant Long-Lasting Improvement after Switch to Incobotulinum Toxin in Cervical Dystonia Patients with Secondary Treatment Failure

**DOI:** 10.3390/toxins14010044

**Published:** 2022-01-06

**Authors:** Harald Hefter, Beyza Ürer, Raphaela Brauns, Dietmar Rosenthal, Sven G. Meuth, John-Ih Lee, Philipp Albrecht, Sara Samadzadeh

**Affiliations:** Department of Neurology, Medical Faculty, Heinrich Heine University of Düsseldorf, Moorenstrasse 5, D-40225 Düsseldorf, Germany; beyza.uerer@uni-duesseldorf.de (B.Ü.); montanabrauns@t-online.de (R.B.); dietmar.rosenthal@med.uni-duesseldorf.de (D.R.); svenguenther.meuth@med.uni-duesseldorf.de (S.G.M.); john-ih.lee@med.uni-duesseldorf.de (J.-I.L.); phil.albrecht@gmail.com (P.A.); sara.samadzadeh@yahoo.com (S.S.)

**Keywords:** secondary treatment failure, incobotulinum toxin, neutralizing antibodies, low antigenicity, complex proteins

## Abstract

Under continuous long-term treatment with abo- or onabotulinum toxin type A (BoNT/A), ~10 to 15% of patients with cervical dystonia (CD) will develop neutralizing antibodies and reduced responsiveness over an ~10-year treatment period. Among the botulinum neurotoxin type A preparations so far licensed for CD, incobotulinum toxin A (incoBoNT/A; Xeomin^®^) is the only one without complex proteins. Whether CD patients with treatment failure under abo- or onaBoNT/A may still respond to incoBoNT/A is unknown. In this cross-sectional, retrospective study, 64 CD patients with secondary treatment failure after abo- or onaBoNT/A therapy who were switched to incoBoNT/A were compared to 34 CD patients exclusively treated with incoBoNT/A. The initial clinical severity of CD, best outcome during abo- or onaBoNT/A therapy, severity at the time of switching to incoBoNT/A and severity at recruitment, as well as all corresponding doses, were analyzed. Furthermore, the impact of neutralizing antibodies (NABs) on the long-term outcome of incoBoNT/A therapy was evaluated. Patients significantly improved after the switch to incoBoNT/A (*p* < 0.001) but did not reach the improvement level obtained before the development of partial secondary treatment failure or that of patients who were exclusively treated with incoBoNT/A. No difference between abo- and onaBoNT/A pretreatments or between the long-term outcomes of NAB-positive and NAB-negative patients was found. The present study demonstrates significant long-term improvement after a switch to incoBoNT/A in patients with preceding secondary treatment failure after abo- or onaBoNT/A therapy and confirms the low antigenicity of incoBoNT/A.

## 1. Introduction

Repetitive injections of botulinum neurotoxin type A (BoNT/A) are the treatment of choice for a variety of indications [1]. Because of the traumatic route of application by transdermal injections of BoNT/A, activation of dendritic and B- and T-cells [2,3] and induction of antibodies (ABs) can hardly be avoided during the treatment course [4]. Some ABs reduce the efficacy of BoNT/A [5]. However, the induction of such neutralizing antibodies (NABs) does not inevitably prevent a clinical response [5]. The correlation between NAB titer and clinical outcome is usually weak [4], and the development of complete secondary treatment failure (CSTF) is rare [6,7]. However, partial secondary treatment failure (PSTF) is frequent [8,9,10], may occur early [11] and increases with the dose per session and treatment duration [8,9]. When PSTF has become manifest, the question arises regarding how to continue treatment. An increase in the dose or a switch to rimabotulinumtoxin type B (rimaBoNT/B) may help for a few treatment cycles [12]. Cessation of BoNT/A therapy or performance of deep brain stimulation (DBS) are the main treatment alternatives recommended so far [5,13]. Here, we demonstrate that switching to the complex protein-free incobotulinum neurotoxin type A (incoBoNT/A; Xeomin^®^ [14,15]) is another long-term alternative for patients with PSTF after abobotulinum neurotoxin type A (aboBoNT/A; Dysport^®^) or onabotulinum neurotoxin type A (onaBoNT/A; Botox^®^).

List of abbreviations: aboBoNT/A = abobotulinum neurotoxin type A, ABs = antibodies, ADOS = dose at recruitment, ATSUI = TSUI at recruitment, BDOS = dose at time of best TSUI, BoNT/A = botulinum neurotoxin type A, BTSUI = best TSUI, CD = cervical dystonia, CSTF = complete secondary treatment failure, DBS = deep brain stimulation, DUR = duration of incoBoNT/A therapy, GCP = good clinical practise, IDOS = dose at onset of therapy, IMP = patient’s assessment of the improvement of CD, incoBoNT/A = incobotulinum neurotoxin type A, ITSUI = TSUI at onset of BoNT therapy, MHDA = mouse hemidiaphragm assay, NAB = neutralizing antibodies, onaBoNT/A = onabotulinum neurotoxin type A, PSTF = partial secondary treatment failure, rimaBoNT/B = rimabotulinum neurotoxin type B, SDOS = dose of incoBoNT/A at switch to incoBoNT/A, STF = secondary treatment failure, STSUI = TSUI score at switch to incoBoNT/A, SWI = switcher group, TTB = time to best TSUI, TTS = time to switch, XEO-Mono = group of CD patients being exclusively treated with incoBoNT/A.

## 2. Results

### 2.1. The Staircase-like Improvement of CD with Repetitive Injections Every Three Months

Duration of efficacy of a standard dose of BoNT/A in the BoNT/A treatment of cervical dystonia (CD) usually exceeds 3 months [14]. When a CD patient is reinjected after 3 months, he/she starts from a better situation than before the previous injection [16]. With repetitive injections every 3 months, CD will gradually improve and reach a stable plateau around 50 to 60% improvement or a TSUI score [17] around 4 to 5 [8,18,19].

As a first sign of NAB-induced reduction of efficacy, the duration of efficacy will decrease [5]. This implies that, in case of NAB-induced secondary treatment failure (STF), the stable plateau cannot be maintained, and a secondary gradual worsening occurs (as long as the duration of the treatment cycle is kept constant at 3 months) [11,18].

Therefore, the analysis of the time to best improvement (TTB) and the best outcome (BTSUI) yields an indirect hint whether a patient has developed NABs or not [19]. We compare the initial severity of CD (ITSUI) and initial dose (IDOS) at onset of BoNT/A therapy with the best outcome (BTSUI) and dose at best outcome (BDOS). We also present data regarding the extent to which the severity of CD worsened (STSUI) and the dose of incoBoNT/A to which the patient was switched. These treatment-related data were extracted from the charts of the patients and compared to the severity of CD at recruitment (ATSUI) and the dose of incoBoNT/A used at recruitment (ADOS).

### 2.2. Milestones of Treatment in the Entire Switch Group

The mean AGE (at the onset of BoNT/A therapy) in the switcher SWI group was 46.8 +/− 12.6 years. The female/male ratio was 1.6. To demonstrate that all switchers had responded well at the beginning of BoNT/A therapy (see Section 2.4) the ITSUI score (mean/SD: 8.77/3.44) was compared to the BTSUI score during the course of pretreatment before the switch (mean/SD: 4.04/3.25). BTSUI was highly significantly (*p* < 0.001) lower than ITSUI and occurred after 3.7 +/− 3.6 years (TTB). During the following treatment, a secondary highly significant (*p* < 0.001) worsening was observed in the switchers (see Section 2.4) after a long time period, which varied by individual. Patients were finally switched to incoBoNT/A after 7.7 +/− 5.3 years (TTS) on average. The mean STSUI score was 8.32/3.43, which did not differ significantly from ITSUI. Thereafter, switchers were treated with incoBoNT/A for 7.6 +/− 3.1 years (DUR) on average and presented with a mean actual TSUI on the day of recruitment (ATSUI) of 5.40 +/− 2.92, which was highly significantly (*p* < 0.001) lower than STSUI. An rm-ANOVA revealed a significant difference between the TSUI scores at the four different instances (F = 26.0; *p* < 0.001) and highly significant (*p* < 0.001) differences between all repeated measurements (Greenhouse–Geisser tests; Figure 1A).

In Figure 1B, ITSUI, BTSUI, STSUI and ATSUI are presented for all patients in order to demonstrate the individual variability and consistency of the data. There was a noticeable heterogeneous response to incoBoNT/A: in some patients, the severity of CD further increased, but in most patients, it decreased after the switch to incoBoNT/A (Figure 1B).

Using the severity of CD at the day of switch as 100% reference value, patients assessed a significant (*p* < 0.001) mean improvement (IMP) of 42.5 +/− 23%. Only 16% of the switchers reported a further worsening after the switch, 27% reported an improvement between 0 and 40% and 57% reported an improvement better than 40% in comparison to the severity of CD at the day of switch.

In 23% of the switchers, ATSUI was worse compared to STSUI; in 36%, the improvement of the TSUI score ranged between 0 and 40%; and in 41% of the patients, improvement of the TSUI score was better than 40%. The mean improvement of the TSUI score ((STSUI-ATSUI)/STSUI*100) was 35.1 +/− 28.4 %.

In 16 of 59 (= 26.7%) switchers, the mouse hemidiaphragm assay (MHDA test) was positive. The Chi2-test did not detect a difference in the frequency of occurrence of MHDA positive tests between the ABO- and ONA subgroups (Table 1).

### 2.3. Treatment of the XEO-Mono Group

The mean age (at onset of incoBoNT/A therapy) in the XEO-Mono group (see Section 5) was slightly higher than that in the SWI group. The female/male ratio was 1.3, but chi2 testing did not detect a significant difference from the SWI group. ITSUI in the XEO-Mono group was 7.83/3.25, which was not significantly lower than that in the SWI group and was close to the STSUI in the ONA and ABO groups (Table 1). BTSUI was 1.71 +/− 1.77 and occurred after 2.52 +/− 1.86 years. The mean ATSUI was 3.27 +/− 2.35. The mean improvement in the TSUI score ((STSUI-ATSUI)/STSUI*100) was 58.3 +/− 24%, and the mean IMP was 70.2 +/− 28%. All patients in the XEO-Mono group had a negative MHDA test (Table 1).

### 2.4. Comparison of abo- and onaBoNT/A Pretreated Switchers with the XEO-Mono Group

In Figure 2A, the mean ITSUI, BTSUI, STSUI and ATSUI of the ONA- (dark gray bars) and ABO-pretreated groups (light gray bars) are presented and compared to ITSUI, BTSUI and ATSUI of the XEO-Mono group (open bars). A three-group rm-ANOVA did not reveal any differences between the ONA- and ABO-pretreated switchers. This was also true for the doses (IDOS, BDOS, SDOS, ADOS; see Figure 2B and Table 1) as long as a conversion ratio of 3:1 between abo- and onaBoNT/A and a conversion ratio of 1:1 between onaBoNT/A and incoBoNT/A was used, following a European consensus recommendation [20]. Furthermore, the patients’ ratings of the improvement after switching to incoBoNT/A (IMP) did not differ between the ABO and ONA groups (see Table 1).

The three-group rm-ANOVA revealed significant differences between the ABO/ONA groups and the XEO-Mono group for BTSUI (*p* < 0.005), ATSUI (*p* < 0.001) and IMP (*p* < 0.001; Table 1). The comparison of BTSUI between the small ONA group and the XEO-Mono group did not reach significance, but the comparison between BTSUI of the ABO and XEO-Mono groups was significant (*p* < 0.005). The comparison of ATSUI between the ONA- and the XEO-Mono groups (*p* < 0.001) as well as between the ABO- and the XEO-Mono groups (*p* < 0.008) was highly significant. The duration of incoBoNT/A treatment (DUR) was significantly longer (*p* < 0.005) in the ONA group compared to the XEO-Mono group but not in the ABO group. The patient assessment of improvement after onset of incoBoNT/A therapy was significantly higher in the XEO-Mono group than in the ONA (*p* < 0.05) and ABO groups (*p* < 0.001).

In all three treatment groups (ONA, ABO and XEO-Mono group), the mean dose of incoBoNT/A (ADOS) was highly significantly (*p* < 0.001) increased since the onset of incoBoNT/A therapy: by 125 U in the ONA group, by 84 U in the ABO group and by 116 U in the XEO-Mono group (see Table 1 and Figure 2B). We did not observe a significant difference between ADOS in the ONA, ABO and XEO-Mono groups (*p* = 0.057).

### 2.5. Comparison of the NAB-pos and NAB-neg Switchers with the XEO-Mono Group

We only identified significant differences between the NAB-pos and NAB-neg groups for SDOS and IMP (see Table 2). SDOS in the NAB-neg group was significantly lower (*p* < 0.023) than that in the NAB-pos group. Patients with a negative MHDA test experienced a significantly (*p* < 0.009) better outcome (IMP) after incoBoNT/A therapy than MHDA-positive patients with NABs.

We identified significant differences for TTB, ATSUI, ADOS and IMP when the NAB-pos and NAB-neg groups were compared with the XEO-Mono group. TTB was significantly longer in the NAB-pos group than in the XEO-Mono group but not in the NAB-neg group (see Table 2). The difference in ATSUI between the NAB-pos and the XEO-Mono group (*p* < 0.001) and the NAB-neg and the XEO-Mono group (*p* < 0.012) was significant. The difference in ADOS between the NAB-pos- and XEO-Mono groups was significant (*p* < 0.013), but not between the NAB-neg and XEO-Mono groups. IMP was significantly better in the XEO-Mono group than in the NAB-pos and NAB-neg groups (Table 2).

### 2.6. Temporal Development of the Outcome of the AK-pos and the AK-neg Groups before and after the Switch to incoBoNT/A

The mean duration of treatment after switching to incoBoNT/A was more than 2700 days (corresponding to 30 treatment cycles) in the NAB-pos and NAB-neg groups. Figure 1A and Table 2 show that, after this time period, ATSUI was significantly lower than STSUI. In Figure 3, the temporal development of the TSUI scores is demonstrated. Figure 3 shows the last eight mean values of the TSUI scores before the switch (S) and those of 24 treatment cycles after the switch for the NAB-pos group (heavy closed line; standard deviation bars in downward direction) and for the NAB-neg group (light dotted line; standard deviation bars in upward direction). By the definition of PSTF in the present study and inclusion criterion (iv), the TSUI score increased before the switch (S). This increase was much steeper in the NAB-pos group than in the NAB-neg group. In both groups, a rapid decrease in the mean TSUI score and clinical improvement was observed during the first four treatment cycles after the switch (see Figure 3), which was more pronounced in the NAB-pos group than in the NAB-neg group. Then, a plateau of approximately 10 treatment cycles followed, with only small changes in the mean TSUI score in both groups. During the subsequent cycles, few changes were observed in the NAB-pos group, whereas there was a clear trend toward further improvement in the NAB-neg group.

## 3. Discussion

In the present study, long-term improved after switch to incoBoNT/A is demonstrated in 64 CD patients with secondary treatment failure (STF) after abo- or onaBoNT/A. This is the largest study on the long-term outcome of patients with STF, with the longest follow-up duration.

### 3.1. Reasons for the Occurrence of STF during BoNT/A Therapy

The number of clinical indications for BoNT/A applications is continuously growing [1,21,22]. For most indications, repetitive injections of botulinum neurotoxin must be applied to maintain a certain level of improvement. These transdermal repetitive injections alert the immune system, activate dendritic cells [2] and induce B- and T-cell responses with the risk of NAB formation and the development of STF [3]. Analysis of B-cells in BoNT/A-treated patients shows that the presence of BoNT/A molecules is detected by the immune system in most long-term treated patients [3]. PSTF may occur in up to 40% of BoNT/A-treated patients [10]. For several indications, it has been reported that STF may occur even after one to three injections [23,24]. In general, high doses per session, long duration of treatment [8,9] and possibly short duration of treatment cycles and number of booster injections [25] are risk factors for the development of STF.

Induction of antibodies and the antigenicity of a BoNT preparation depend on the clostridial protein content of a BoNT/A vial and the neurotoxin load to achieve a clinical response. This differs considerably between the different BoNT/A preparations [15]. The aboBoNT/A preparation (Dysport^®^, Ipsen^®^, Paris, France) and the onaBoNT/A preparation (Botox^®^, Allergan^®^, Irvine, CA, USA) contain clostridial complex proteins that shield the 150 KD large neurotoxin type A molecule during its passage through the acidic milieu of the stomach [26] and support its transmigration through the intestinal epithelial barrier [27] after oral uptake. They do not shield epitopes of the BoNT/A molecule against antibodies after direct injection into a tissue [28,29] but enhance the immune response [30,31].

During the manufacturing process of the incoBoNT/A preparation (Xeomin^®^, Merz Pharmaceuticals^®^, Frankfurt, Germany), not only the clostridial complex proteins but also biologically inactive fragments are removed, and the total clostridial protein content of a vial of 100 U Xeomin^®^ is reduced to 0.44 ng [32], which is much lower than that of the abo- or onaBoNT/A preparation [32]. This low protein content of the incoBoNT/A preparation leads to a low antigenicity of incoBoNT/A in comparison to abo- or onaBoNT/A, which was confirmed in an animal experiment [33] and in a recent cross-sectional study in patients who received long-term treatment exclusively with incoBoNT/A without NAB induction [18,34]. It has even been reported that in patients who have developed NABs under abo- or onaBoNT/A, NAB titers may decline even below the normal range limit when BoNT/A therapy is switched to incoBoNT/A treatment [35] and that some of these patients have a clinical response to 200 U of incoBoNT/A during the first year after the switch [36]. If the reduction in NAB titers goes along with a long-lasting and not simply a transient clinical improvement, as after the switch to rimaBoNT/B [37], switching to incoBoNT/A would be an alternative to DBS for patients with immunoresistance to abo- or onaBoNT/A [12,37,38]. Therefore, the present study was designed.

### 3.2. Significant Long-Lasting Improvement after Switching to incoBoNT/A

In the present study, a typical cohort of 64 patients with idiopathic CD and PSTF after abo- or onaBoNT/A therapy was analyzed. The mean age at onset of symptoms was close to 47 years, with a female/male ratio of 1.6. Patients had responded well to abo- or onaBoNT/A therapy (Figure 1A,B), experienced a significant improvement after onset of BoNT/A treatment and reached their optimal reduction in the severity of CD after a mean of 3 years. Thereafter, a continuous decline in the efficacy of BoNT/A injections was noticed, and after 7.7 years, the mean severity of CD slowly approached a level close to that at the onset of BoNT/A therapy (Figure 1A,B). The patients and treating physicians were increasingly unsatisfied with the efficacy of BoNT/A therapy and looked for other therapeutic options.

All patients in the SWI group had postponed or had decided against DBS and therefore were switched to treatment with increasingly higher doses of incoBoNT/A (see Figure 2B). The treating physicians observed a continuous improvement after the switch (Figure 3) and documented a significant (*p* < 0.001) mean improvement of 35% after 7.6 years of incoBoNT/A treatment. Patients´ subjective mean global assessment of the improvement (IMP) after the switch to incoBoNT/A was 42.5% and matched the physician´s objective scoring. Usually, physicians overestimate the treatment effect compared to that in the patients’ assessments [39].

Responses to incoBoNT/A varied considerably from excellent to no response (Figure 1B). The remaining severity of CD in the SWI group estimated by means of the TSUI score (ATSUI) was higher (5.40) than that in other CD cohorts (4.5 in [40], 4.9 in [8]) treated long-term with abo- or onaBoNT/A [8,40]. Compared to the long-term outcome in the XEO-Mono group, ATSUI in the SWI group was significantly worse. The large variability of the responses to incoBoNT/A in the SWI group is in line with the observation in MHDA-positive CD patients, that NAB titers may increase in some patients but will decrease in the majority of patients with ongoing incoBoNT/A therapy, even below the normal range limit [35,36]. Therefore, it is not surprising that the percentage of MHDA-positive patients was lower than that in other cohorts of CD patients with PSTF, with up to 50% MHDA-positive patients [41], and that the percentage of MHDA-positive patients was much lower in the ONA group (9.1%) (with a mean duration of incoBoNT/A treatment of 8.75 years) than in the ABO group (29.1%) (with a mean duration of treatment of 7.23 years).

### 3.3. Comparison of Switch to incoBoNT/A Treatment and DBS in Patients with PSTF

However, after DBS, a large variability of responses can also be observed [42]. The DBS of CD patients leads to a significant improvement [13,42], though not in all cases, and is associated with a considerable percentage of side effects [13]. Even the induction of a severe parkinsonian syndrome after DBS in patients with CD has been reported by different teams performing DBS operations [43,44].

The results of the present study are difficult to compare to results after DBS. However, in the study by Schönecker et al. [42], improvement after 76 DBSs in CD patients was quantified using the TWSTRS severity score [45], which allows to some extent a comparison of the relative TSUI score improvement of the SWI group with the relative TWSTRS improvement due to DBS-operation. After DBS in less than 10% of the patients, a worsening was observed, but only one-third of the patients had an improvement better than 40% [42]. In the present study, 23% of the patients presented with no change or with worsening of the TSUI score, but 41% presented with an improvement larger than 40%. In both studies, the mean relative improvement was close to 35%.

In our experience, DBS is superior to BoNT/A injections in patients with a pure antecollis or antecaput [18], whereas an excellent outcome can be achieved in CD patients with a retrocaput component by injection of the deep neck muscles [46]. Regardless, BoNT/A therapy should be optimized by injection according to the cap/col concept [46] and inclusion of the deep neck muscles into the injection pattern.

### 3.4. Comparison of incoBoNT/A and rimaBoNT/B Injection Therapy in Patients with PSTF

RimaBoNT/B injections are effective in patients with an STF after abo- or onaBoNT/A. This has been confirmed by several studies [37,47]. Side effects after BoNT/B were more frequent than those after BoNT/A but were tolerable in most cases [37,47]. In our center, the outcome after the first two aboBoNT/A injections and the first two rimaBoNT/B injections was nearly identical, even when low doses of rimaBoNT/B were used, but no correlation between the response to BoNT/A and BoNT/B was found. This implies that the response to BoNT/B cannot be predicted from previous BoNT/A injections [37]. However, after the third, and much more pronounced after the fourth rimaBoNT/B injection, the clinical response started to decline, and 3 out of 17 patients developed a second PSTF [36]. In another study analyzing BoNT/B treatment with higher doses of rimaBoNT/B in 10 CD patients with STF after BoNT/A, a second STF occurred in 6 out of 9 (67%) patients after four rimaBoNT/B injections [12].

### 3.5. No significant Difference between Pretreatment with abo- or onaBoNT/A

When switchers were split up into those patients who started their BoNT/A with onaBoNT/A (ONA group) and those who started with aboBoNT/A (ABO group), no difference in outcome was found between these two subgroups. BTSUI, STSUI, TTS and ATSUI did not differ. As long as a conversion ratio between onaBoNT/A and aboBoNT/A units of 1:3 (following a European consensus recommendation [20]) was used, no significant difference in any of the dose parameters (IDOS, BDOS, SDOS, ADOS) was found. The ratio of IDOS in our cohort was 210:620 = 1:2.95 and thus was very close to the 1:3 ratio in the consensus paper [20]. The ratio of BDOS was 321:843 = 1:2.63, which is closer to the value of 1:2.5 that is preferred by others (for details, see [20]).

The chi2-test comparing the presence of NABs in the ONA and ABO groups failed to be significant, mainly because of the small size of the ONA group. A precise analysis of differences in NAB prevalence must be based on a detailed Kaplan–Meier analysis [9,18].

### 3.6. Worse Outcome in Switchers Compared to the XEO-Mono Group

BTSUI in the XEO-Mono group was significantly lower than BTSUI during pretreatment with abo- or onaBoNT/A (see Table 1). The best TSUI score improvement ((ITSUI-BTSUI)/ITSUI) in the ONA group was 57.5%, in the ABO group was 56% and in the XEO group was 78%. In most of the switchers (63%), the response to incoBoNT/A did not reach the same level of improvement as the best response during pretreatment. This implies that the development of STF should be avoided [18] and that the most purified BoNT/A preparation [18,48] should be used from the very beginning of BoNT/A therapy, as recommended by Aoki and Guyer [48]. It has been demonstrated that the development of a PSTF may occur early in the course of BoNT/A treatment [11]. The significantly higher BTSUI in the ONA and ABO group in comparison to the BTSUI in the XEO-Mono-group is consistent with this observation.

### 3.7. Tendency toward a Better Outcome in NAB-Negative Compared to NAB-Positive Patients

It has been reported that in patients with NABs who were switched to incoBoNT/A treatment, NAB titers were not boosted in most of these switchers. In more than 50% of the patients, the titers progressively declined over a period of 4 years, even below the detection limit of the MHDA [35]. Patients in the present study were tested for the presence of NABs after a mean of 7 years following the switch to incoBoNT/A. Therefore, those patients with persistently elevated NAB titers form a negatively selected subgroup. Analysis of the dose and the physician´s assessment of the outcome did not reveal significant differences between the NAB-pos and the NAB-neg groups. However, the patients´ global assessment of improvement (IMP) revealed a significantly worse outcome in the NAB-pos group compared to the NAB-neg group. Possibly, in some of the NAB-neg patients, NAB titers had declined [35,36], and the improvement progressively increased, as indicated in Figure 3. However, the correlation between NAB titer and clinical response is complex [4,36] and should be analyzed in more detail.

## 4. Conclusions

In the present study, a significant, long-lasting improvement was demonstrated in patients with PSTF after abo- or onaBoNT/A therapy when switched to incoBoNT/A. Therefore, switching to incoBoNT/A is a relevant alternative to DBS in patients with STF after abo- or onaBoNT/A therapy. However, the improvement after incoBoNT/A in patients with STF did not reach the level of improvement observed in patients with CD who were exclusively treated with incoBoNT/A. Therefore, it is recommended to use incoBoNT/A from the very beginning of BoNT/A treatment to reduce the risk of development of PSTF and antibody induction [18] and to achieve an optimal long-term outcome of BoNT/A therapy.

### Limitations of the Present Study and Recommendation for Further Studies in the Future

The results of the present study would have been more convincing if antibodies had repeatedly been tested in parallel to demonstrate that the improvement observed in clinical practice goes along with a reduction in NAB titers. However, this approach would have by far exceeded the cost limits of the present study. Therefore, the performance of a multicenter study with careful clinical investigation on the one hand and continuous NAB testing on the other hand is recommended, which is a challenging study design. Furthermore, there is a need for a randomized study that carefully controls the type of CD, side effects and outcomes to compare DBS and incoBoNT/A injection therapy in patients with PSTF after abo- or onaBoNT/A therapy.

## 5. Methods

All patients gave written informed consent.

### 5.1. Patients and Treatment-Related Data

In the present study, only patients with CD were analyzed for 3 different reasons: (1) CD is the most frequent focal dystonia [49]; (2) in our institution, the severity of CD is scored before each injection using the TSUI score [16]; and (3) we use a well-defined definition for STF in CD (see inclusion criteria (4) and [11] for details).

For the switchers (SWI group) inclusion criteria were (1) age >= 18 years, (2) diagnosis of idiopathic CD, (3) onset of BoNT therapy in our outpatient clinic, (4-a) satisfactory pretreatment with abo- or onaBoNT/A according to patient assessment and with a response to abo- or onaBoNT/An of at least 3 TSUI score points (physician´s assessment), (4-b) confirmed systematic worsening of the TSUI score over 2 treatment cycles of at least 3 TSUI score points before the switch to incoBoNT/A and lack of symptom reduction reported by the patient during these 2 cycles and (5) continuous incoBoNT/A treatment. The exclusion criteria were (1) presence of a pure antecollis or antecaput (for details, see [50]) and (2) interruption of the BoNT therapy for more than one treatment cycle.

For the patients who had exclusively been treated with incoBoNT/A (XEO-Mono group), the inclusion criteria and exclusion criteria were the same as those for the SWI group, with the exception of the inclusion criterium (4).

Patients were recruited between 1 January 2016 and 31 December 2017. During these 2 years, 64 switchers and 34 CD patients who were exclusively treated with Xeomin^®^ were recruited.

### 5.2. Determination of Neutralizing Antibodies

On the day of recruitment, blood samples were taken and deep frozen until all patients were included. Then, blood samples were coded and sent off to a blinded contractor (Toxogen^®^ GmbH, Hannover, Germany) to be analyzed by means of the mouse hemidiaphragm assay (MHDA; [51]) for the purpose of neutralizing antibodies in one batch. For each sample, the paralysis time was determined, which is the outcome measure of the MHDA [51]. When a paralysis time exceeded 60 min, the MHDA test was classified as positive. A complete list of paralysis times (*n* = 94) and whether a blood sample was classified as MHDA-positive or not was returned to our institution. For 4 switchers, blood samples could not be analyzed: blood samples of 2 switchers were lost, and 2 samples did not contain enough material for the performance of the MHDA. These patients were not included in the final analysis. For 1 MHDA-positive patient, it was unclear whether he had initially been treated with abo- or onaBoNT/A.

### 5.3. Outcome Measures

The primary (objective) outcome measure was the TSUI score at the day of recruitment (ATSUI). In our clinic, the TSUI score is always determined at the end of a treatment cycle, just before the next injection is applied. The patient’s global assessment of the remaining severity of CD (PGA) as a percentage of the CD severity at the onset of incoBoNT/A therapy was used as a secondary (subjective) outcome measure. The difference to 100% was the patient´s global assessment of the improvement of CD since the onset of incoBoNT/A therapy (IMP). Patients were familiar with this assessment procedure because they had to assess the remaining severity of CD before each BoNT injection.

Treatment related at the onset of BoNT/A therapy (initial TSUI score (ITSUI); initial dose (IDOS), best TSUI score (BTSUI) and dose (BDOS) at the time of best TSUI score, TSUI score at the switch to incoBoNT/A (STSUI) and the incoBoNT/A dose at switch (SDOS) were extracted from the charts of the patients. Duration of incoBoNT/A therapy (DUR), severity of CD at recruitment (ATSUI) and dose of incoBoNT/A at recruitment (ADOS) were documented by the treating physician. For the sake of comparison, BoNT/A doses were converted into unified dose units (uDU) by leaving inco- and onaBoNT/A doses unchanged and dividing aboBoNT/A doses by 3. This conversion ratio has been used in a previous meta-analysis on neutralizing antibodies in botulinum toxin therapy [52] and has been discussed in a European consensus paper [20].

### 5.4. Statistics

Switchers (SWI group) were split according to pretreatment: the ABO group comprised all patients who had initially been treated with aboBoNT/A, and the ONA group comprised those with initial onaBoNT/A treatment. Furthermore, switchers were also subdivided into an NAB-pos group containing all MHDA-positive patients and an NAB-neg group comprising all MHDA-negative patients.

A three-group repeated measurements rm-ANOVA was calculated to detect differences between the ABO-, ONA- and XEO-Mono groups (Table 1). To detect differences between measurements, Greenhouse–Geisser subtests were used.

Furthermore, a second three-group rm-ANOVA was performed to compare the NAB-pos, NAB-neg and XEO-Mono groups (Table 2). A Chi2-test was used to compare the female/male ratio in the SWI and XEO-Mono group and the percentage of MHDA positive patients in ABO- and ONA subgroups. All tests were part of the SPSS^®^ statistics package (version 25; IBM, Armonk, NY, USA).

## Figures and Tables

**Figure 1 toxins-14-00044-f001:**
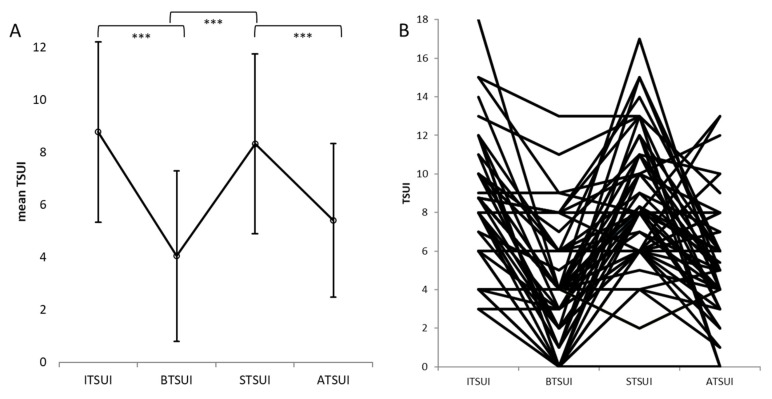
(**A**) Mean values and standard deviations of the initial TSUI scores (ITSUI), the best TSUI scores (BTSUI), the TSUI scores on the day of switching to incoBoNT/A (STSUI) and the actual TSUI scores on the day of recruitment (ATSUI) in the SWI group. The comparison between ITSUI and BTSUI reveals a highly significant improvement of CD, the difference between BTSUI and STSUI reveals a highly significant second worsening and the difference between STSUI and ATSUI shows a second significant improvement after the switch to incoBoNT/A. (**B**) The individual data of ITSUI, BTSUI, STSUI and ATSUI in the SWI group underlying the mean values and standard deviations are presented to demonstrate consistency and variability of the individual data. (*** The comparison between two groups are highly significant).

**Figure 2 toxins-14-00044-f002:**
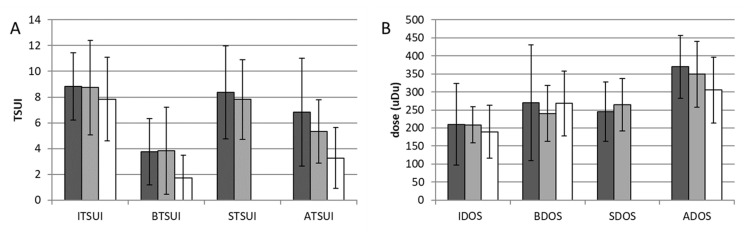
(**A**) Mean ITSUI, BTSUI, STSUI and ATSUI of the ONA group (dark gray bars) and ABO group (light gray bars) are to mean ITSUI, BTSUI and ATSUI of the XEO-Mono group (open bars). (**B**) Mean IDOS, BDOS, SDOS and ADOS of the ONA group (dark gray bars) and ABO group (light gray bars) are compared to IDOS, BDOS and ADOS of the XEO-Mono group (open bars) after conversion to unified dose units (uDU (for details see Section 5)).

**Figure 3 toxins-14-00044-f003:**
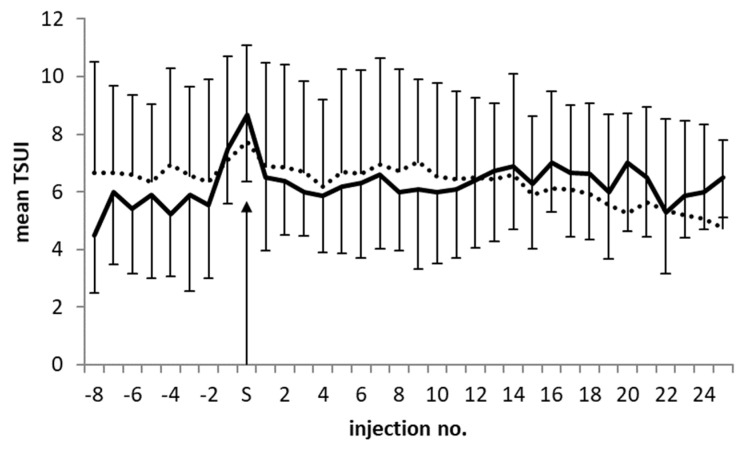
Temporal development of the mean TSUI scores of 8 treatment cycles before the switch and 24 treatment cycles after the switch in NAB-positive switchers (heavy line and error bars extending downward) and NAB-negative switchers (dotted line and error bars extending upward).

**Table 1 toxins-14-00044-t001:** Comparison of the outcomes in the ONA, ABO and XEO-Mono groups.

Parameter	ONA Group	ABO Group	XEO-Mono Group	*p*-Value ONA/ABO	*p*-Value ONA/XEO	*p*-Value ABO/XEO
*n*=	11	48	34			
NAB-pos	1 (9.1%)	14 (29.1%)	0 (0%)	n.s.	n.s.	0.05
female/male	6/5	31/17	19/15	n.s.	n.s.	n.s.
age at onset (years)	44.93/8.01	47.25/13.69	50.98/12.19	n.s.	n.s.	n.s.
ITSUI	8.83/2.61	8.74/3.67	7.84/3.25	n.s.	n.s.	n.s
IDOS (U)	210/113 (Botox^®^)	620/150 (Dysport^®^)	189/73 (Xeomin^®^)	n.s.	n.s.	n.s.
BTSUI	3.75/2.59	3.83/3.36	1.71/1.77	n.s.	n.s.	0.005
BDOS (U)	321/229 (Botox^®^)	843/697 (Dysport^®^)	267/90 (Xeomin^®^)	n.s.	n.s.	n.s.
TTB (days)	1085/739	1354/1362	920/679	n.s.	n.s.	n.s.
STSUI	8.36/3.60	7,82/3.09	n.a.	n.s.	n.a.	n.a.
SDOS (uDU)	245/82 (Xeomin^®^)	265/73 (Xeomin^®^)	n.a.	n.s.	n.a.	n.a.
TTS (days)	2639/1540	2849/2015	n.a.	n.s.	n.a.	n.a.
ATSUI	6.82/4.20	5.32/2.46	3.27/2.35	n.s.	0.001	0.008
ADOS (uDU)	370/87 (Xeomin^®^)	349/92 (Xeomin^®^)	305/91 (Xeomin^®^)	n.s.	n.s.	n.s.
DUR (days)	3150/984	2604/1113	2283/844	n.s.	0.005	n.s.
IMP (%)	52.3/25.6	39.6/37.7	70.2/22	n.s.	0.05	0.001

NAB-pos = number and percentage of MHDA-positive patients; for the definition of ITSUI, IDOS, BTSUI, BDOS, TTB, STSUI, SDOS, TTS, ATSUI, ADOS, DUR and IMP, see Methods Section 5.3.

**Table 2 toxins-14-00044-t002:** Comparison of the outcomes in the NAB-pos, NAB-neg and XEO-Mono groups.

Parameter	NAB-pos Group	NAB-neg Group	XEO-Mono Group	*p*-Value NAB-pos/ NAB-neg	*p*-Value NAB-Pos/ XEO	*p*-Value NAB-neg/ XEO
*n*=	16	44	34			
female/male	11/5	26/18	19/15	n.s.	n.s.	n.s.
age at onset	44.98/11.15	47.79/8.75	50.98/12.19	n.s.	n.s.	n.s.
ITSUI	9.86/3.66	8.18/3.27	7.84/3.25	n.s.	n.s.	n.s.
IDOS (uDU)	216/70	211/76	189/73 (Xeomin^®^)	n.s.	n.s.	n.s.
BTSUI	4.14/3.07	3.81/3.34	1.71/1.77	n.s.	n.s.	n.s.
BDOS (uDU)	247/72	252/117	267/90 (Xeomin^®^)	n.s.	n.s.	n.s.
TTB (days)	1707/1410	1197/1211	920/679	n.s.	0.005	n.s.
STSUI	8.19/2.63	8.00/3.57	n.a.	n.s.	n.a.	n.a.
SDOS (uDU)	294/66	243/74	n.a.	0.023	n.a.	n.a.
TTS (days)	3014/1762	2722/1960	n.a.		n.a.	n.a.
ATSUI	6.44/2.00	5.29/3.28	3.27/2.35	n.s.	0.001	0.012
ADOS (uDU)	384/82	338/91	305/91 (Xeomin^®^)	n.s.	0.013	n.s.
DUR (days)	2633/1103	2822/1130	2283/844	n.s.	n.s.	n.s.
IMP (%)	18/43	54/23	70.2/22	0.009	0.001	0.05

For the definition of ITSUI, IDOS, BTSUI, BDOS, TTB, STSUI, SDOS, TTS, ATSUI, ADOS, DUR and IMP, see Method Section 5.3.

## Data Availability

Data available on request due to restrictions, e.g., privacy or ethical. The data presented in this study are available on request from the corresponding author.

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
