# Peer review of "Significant Long-Lasting Improvement after Switch to Incobotulinum Toxin in Cervical Dystonia Patients with Secondary Treatment Failure"

_toxins, 2022, doi:10.3390/toxins14010044_

Round 1
Reviewer 1 Report
Authors investigated patients with cervical dystonia with secondary unresponsiveness to abo (Dysport) and ona (Botox)-botulinum toxin, and their long-term responsiveness after switching to inco (Xeomin) -botulinum toxin (patients responded favorably, though not as good as their initial response).
The manuscript discusses an important practical issue supported by additional clinically important subjects like the presence of neutralizing antibodies and comparison the results with that of pallidal DBS.
The problem with the manuscript is extensive use of abbreviations which are explained at Part 5 (Methods), after numerous use of the abbreviated forms before their explanation. Abbreviations should be explained when they are first written in the text.
I could not understand whether placing the Methods section as the last section after Discussion is an unusual rule of the journal or is a mistake.
Author Response
Reviewer 1 is absolutely right:
In the “Toxins” journal Methods are placed at the end of the manuscript.
We therefore do not only list the abbreviations after the introduction, but also give a short explanation.
We also take care that abbreviations are explained when they are first written in the text.
Reviewer 2 Report
- Although I find the case report in section 2.1 of the Results, this does not seem to add much with regards to the rest of the study. I would consider removing it.
- Inconsistent reporting baseline parameters in the different analysis sections. Could be improved.
- Figure 2B is very difficult to interpret. I suggest presenting the results as mean changes from baseline, though preferably accounting for the baseline Tsui scores.
Author Response
This case report was included, to demonstrate in detail the improvement in a patient with complete secondary treatment failure after aboBoNT/A when he was switch to incoBoNT/A.
Since two reviewers have recommended to remove this case, we follow their advice.
We checked the baseline parameters. Why numbers may slightly vary is explained in the methods section.
In Fig.2B the individual data are presented underlying the mean values and standard deviations in Fig. 2A. If we had used changes of TSUI-scores it would have been impossible to demonstrate convincingly that in the majority of patients severity of CD worsens again until it approaches nearly the same plateau as before onset of BoNT therapy. We therefore prefer to leave the figure as it is but have modified the figure legend.
Reviewer 3 Report
please find detail in the word

Author Response
Reviewer 3 is absolutely right:
We now explain the abbreviations in more detail including U and uDU.
The single case is removed in the revised manuscript.
We have presented the date as they are. But Reviewer 3 raises a very interesting question: how to determine the BT dose at each therapy? Is it not dependent on the severity of CD?
Our answer to this question is: definitely no. At least not in our institution as well as in many centers in Germany.
Here is the simple reason: If BT therapy is started with a given dose and the patient improves we do not reduce the dose but maintain the dose.
To condense the article we have removed the case report and have shortened the discussion.
This seems to be a misunderstanding of what we had in mind. We did not mean that patients were recruited under continuous incoBoNT/A treatment ..of more than one treatment cycle. We wanted to say that we recruited only patients who did not interrupt there BoNT treatment for longer than 1 treatment cycle.
Inclusion criterion (v) is improved now.
98% of the patients had been treated with incoBoNT/A for longer than 1.4 years. (see Results).
Reviewer 3 is absolutely right.
In the original version of the manuscript a detailed legend to figure 3A and to figure 3B is present explaining in detail that dark gray bars correspond to the ONA-group, light gray bars to the ABO-group and open bars to the XEO-Mono-group.
To improve understanding, we have modified the legend to Fig. 3 and 4.

Reviewer 4 Report
This paper is a retrospective analysis who switched from abo- or onabotulinumtoxinA to incobotulinumtoxinA, claiming that there was significant improvement after switching. This study has major problems at least in the conclusion as follows:
- This is not controlled by a group who continued to use abo- or ona-BoNTs for the same long period as the switchers.
- The rating should be TWSTRS rather than old TSUI score.
- The conclusion is not supported by the present data.
'In the present study, a significant, long-lasting improvement was demonstrated in 343 patients with PSTF after abo- or onaBoNT/A therapy when switched to incoBoNT/A. '
- It is possible that the improvement due to the placebo effects or natural course.
'Therefore, switching to incoBoNT/A is a relevant alternative to DBS in patients with STF 345 after abo- or onaBoNT/A therapy.'- This is not justified by the present data.
- 'However, the improvement after incoBoNT/A in patients with STF did not reach the level of improvement observed in patients with CD who were exclusively treated with incoBoNT/A. Therefore, it is recommended to use incoBoNT/A from the very beginning of BoNT/A treatment to reduce the risk of development of PSTF and antibody induction [34] and to achieve an optimal long-term outcome of BoNT/A therapy.
- This is not scientifically supported.
Author Response
The cohort of switchers in the present study is the largest cohort of switchers presented so far with a mean duration of 7.6 years of incoBoNT/A treatment.
Knowing that switching to incoBoNT/A might improve severity of CD we think it is unethical not to switch to incoBoNT/A and maintain the BoNT preparation under which patients had experienced a severe secondary worsening.
Reviewer 4 is right: TWSTRS would have been better. But this is not a longitudinal study designed over 15 years, but a retrospective analysis of data being produced during routine BoNT treatment. In most centers neither TWSTRS nor TSUI-score are used in daily practice. This has nothing to be with old and new but with availability of resources to perform tests.
All sentences in the conclusion are supported by the presented data. We even digitized the published data of Schönecker et al. 2015 and reanalyzed their data, to support the sentence: Therefore, switching to incoBoNT/A is a relevant alternative to DBS in patients with STF after abo- or onaBoNT/A therapy.
No placebo is given. We have studied and commented on the natural course of CD. Relapses are rare in CD (see Hefter et al. Disease progression.. Front Neurol doi:10.3389/fneur.2020.588395). Therefore, the systematic improvement and secondary worsening after abo- or onaBoNT therapy and second improvement after incoBoNT/A can never be the result of the natural course of CD.
This is discussed in detail in the discussion. (see comment above). It will be further analyzed by a randomized comparative study which is on the way.
This is clearly demonstrated in Fig. 3A and in Table 2 and 3.
So far, no patient has been reported to have developed NAB induced STF under incoBoNT/A monotherapy.
Therefore, we think that our recommendations are based on solid scientific data.
Round 2
Reviewer 4 Report
This paper has been improved, and the nature of this retrospective study was well understood.